# Effect of Hydrogen on AM Pyroptosis Induced by Severe Burns in Rats

**DOI:** 10.3390/jpm13030377

**Published:** 2023-02-21

**Authors:** Ning Luo, Hua Lin, Linlin Zhang, Yi Jiang, Yue Zhao, Qingqing Han, Xin Wang, Yonghao Yu, Chao Qin

**Affiliations:** 1Department of Anesthesia, Tianjin Medical University General Hospital, Heping District, Tianjin 300052, China; 2Tianjin Institute of Anesthesiology, Tianjin 300052, China

**Keywords:** severe burns, acute lung injury, hydrogen, pyroptosis, NLRP3, GSDMD

## Abstract

Background: Hydrogen has anti-inflammatory and antioxidant effects and is beneficial to multiple organs. However, its effect on alveolar macrophage (AM) pyroptosis induced by burns is still unclear. The purpose of this research was to study the possible positive effects of hydrogen on burn-induced lung injury and the effects of hydrogen on AM pyroptosis during acute lung injury (ALI) induced by burns. Methods: In this study, histological changes in rat lungs in vivo were evaluated by micro-CT, and histological changes in isolated lungs were evaluated by hematoxylin and eosin (HE) staining. The expressions of leucine rich repeat (LRR) and pyrin domain (PYD) containing protein 3 (NLRP3), caspase-1 and Gasdermin-D (GSDMD) were analyzed by Western blotting. The expression of GSDMD was measured by immunofluorescence to evaluate the levels of lung inflammation and pyroptosis. The level of inflammation was assessed by enzyme-linked immunosorbent assay (ELISA). Pyroptosis was observed by transmission electron microscopy. Results: We observed that severe burn resulted in increased IL-1β and IL-18, overexpression of NLRP3 and caspase-1 proteins, and pyroptosis in rat lung tissues, as demonstrated by GSDMD overexpression and electron microscopy of AMs. We also observed that hydrogen treatment partially reversed the increase in lung tissue density and reduced pulmonary inflammation. Moreover, hydrogen reduced the HE pathological injury score in the lung tissues of severely burned rats. Hydrogen treatment significantly reduced the contents of IL-1β and IL-18 in the lung tissues and decreased the expression of NLRP3, caspase-1 and GSDMD proteins compared with the burn group. Transmission electron microscopy results also showed that the number of AM membrane pores was significantly reduced in the hydrogen treatment group. Conclusions: The results of this study suggest that hydrogen may protect against ALI induced by burn injury by inhibiting pyroptosis of macrophages via NLRP3.

## 1. Background

In the past 20 years, burns have become an important public health problem, and more than 250,000 people die from burns every year [1]. Large burn areas are accompanied by a strong inflammatory response, which most often leads to systemic inflammatory response syndrome (SIRS). SIRS is often accompanied by multiple organ dysfunction syndrome (MODS). The lung is the sentinel organ in MODS caused by burns, and ALI or acute respiratory distress syndrome (ARDS) can be life-threatening [2]. Therefore, early intervention and treatment of ALI are very important for improving the prognosis of burn patients.

Alveolar macrophages (AMs) participate in the immune response during ALI [3]. Under the condition of inflammation, AMs in BALF account for 90% of the total cell population [4], indicating that AMs play an important role in the occurrence and development of ALI. The inflammatory body is a polyprotein complex that plays a key role in innate immunity. It can recognize endogenous, noninfectious, and damage-associated molecular patterns (DAMPs) in addition to exogenous pathogen-associated molecular patterns (PAMPs) [5]. NLRP3 is the most well-studied inflammasome and the only one that identifies DAMPs released after burns, and it has been proven to be related to ALI. When NLRP3 is injured by pathogens or sterile tissue or activated by metabolic stress, caspase-1 and the cytokines IL-1β and IL18 are secreted, which leads to persistent inflammatory reactions and pyroptosis [6].

Pyroptosis is a process of inflammatory cell demise. The main function of pyroptosis is to cause strong inflammatory reactions that protect the host from microbial infection. However, excessive pyroptosis can lead to sepsis, autoimmune disorders and other inflammatory diseases [7]. Pyroptosis may be typical or atypical. Among them, the typical pathway responds to PAMPs and DAMPs. In the typical pathway, GSDMD is cleaved by caspase-1 into the N-terminus and C-terminus, and the N-terminus of GSDMD forms a transmembrane pore. It can release IL-1β, IL-18 and other cytokines. Moreover, it can interfere with the adjustments of ions and water, leading to intense inflammation and pyroptosis [7]. AM pyroptosis aggravates pulmonary inflammation [8].

Molecular hydrogen has many biological effects, such as anti-inflammation, antioxidation, anti-apoptosis, anti-shock, and autophagy regulation [9]. Hydrogen, hydrogen saline and hydrogen water reduce the levels of proinflammatory cytokines to suppress inflammation. Due to its powerful efficacy and novelty, H_2_ has great potential in the prophylaxis and treatment of many illnesses [10]. Moreover, the protective effect of hydrogen has been reported in many diseases, including heart muscle, brain, lung, liver, kidney and intestinal tract diseases [11,12,13,14,15]. The seventh edition of Chinese Clinical Guidance for COVID-19 Pneumonia Diagnosis and Treatment issued by the China National Health Commission recommends the inhalation of O_2_ mixed with H_2_. The recommendation recognized the efficacy of hydrogen in the treatment of respiratory diseases. A study involving 41 patients with NCP showed that hydrogen can downregulate the level of inflammatory factors including IL-2, IL-7, IL-10, and TNF-α NE [16].

In summary, we propose that hydrogen has a protective effect on burn-induced ALI, and the mechanism may be related to its inhibition of the NLRP3 signaling pathway. Therefore, we aimed to confirm this hypothesis in the burn-induced rat ALI model and found for the first time that hydrogen ameliorates the development of burn-induced ALI by inhibiting the NLRP3/caspase-1/GSDMD signaling pathway, which may provide a new direction for the treatment of burn-induced ALI.

## 2. Materials and Methods

### 2.1. Animals and Experimental Groups

Healthy adult male Sprague-Dawley rats (200–230 g; 6–7 weeks old) were acquired from Huafukang Bioscience Company (license number SCXK 2019-0008; Beijing, China). The number of rats in each group was 28, and the total number of rats used in the experiment (plus the rats that died due to modeling) was 120. The rats were randomly fed a standard diet and tap water at 23 to 25 °C and housed under conditions of 55 to 60% humidity with a light/dark cycle of 12 h. The animals were exposed to the environment for at least 1 week before the experiment. The sham operation (sham), sham operation plus hydrogen (sham + H_2_), severe burn (burn), and severe burn plus hydrogen (burn + H_2_) groups were randomly established.

### 2.2. Burn Model

Twenty-four hours before the severe burn model was established, the backs of the rats were depilated with an electric depilator, and the rats were fasted 12 h before modelling. The rats were anesthetized with sevoflurane (5%) mixed with oxygen (the gas flow rate was 1.5 L/min). The ZS-YLS-5Q desktop super temperature-controlled scald instrument (ZS-YLS-5Q, Beijing ZhongShiDiChuang Technology Development Company, Beijing, China) was used to scald the backs of the rats at 94 °C for 18 s in a 4 cm × 4 cm area. The model of a 40% total body surface area III degree burn was prepared. The rats in the sham and sham + H_2_ groups were exposed to skin temperature only for back depilation. Immediately after injury, 5 mL of 0.9% physiological saline was injected intraperitoneally. In a sealed plexiglass box with an inlet and an outlet, a hydrogen generator was used to generate hydrogen, and the hydrogen and air were mixed by a gas flowmeter and entered the box at a rate of 4 L/min. The oxygen volume fraction in the box was 21%, and the hydrogen volume fraction was maintained at 2% with a hydrogen detector. A layer of calcium lime at the bottom of the box was used to absorb the carbon dioxide produced by the rats during breathing. At 1 and 6 h post-injury, rats in the sham + H_2_ and burn + H_2_ groups inhaled hydrogen in this environment for 1 h, while rats in the sham and burn groups were in the same environment and inhaled air. Throughout the entire process, the volume fraction of hydrogen in the box was continuously monitored, and the air circulation in the experimental environment was monitored to avoid hydrogen accumulation accidents. In the course of the experiment, the ambient temperature was 23–25 °C, and the rats could freely drink and eat.

### 2.3. Sample Collection

At 12 and 24 h after modeling, six rats in each group were anesthetized with sevoflurane to expose the heart. The right atrial appendage was cut, and a needle tip was inserted into the left cardiac apex and lavaged with PBS, followed by fixation with tissue cell fixation solution. Lung tissue was obtained and put into tissue cell fixation solution. In addition, the hearts of six rats were exposed under the same anesthesia, the right atrial appendage was cut, the needle tip was inserted into the left cardiac apex and lavaged with PBS, and the lung tissue was immediately preserved at −80 °C for later use.

### 2.4. CT Scan and Density Analysis of the Lung Tissue

At 12 and 24 h after modeling, six rats in each group were sedated with chloral hydrate, and their lungs were scanned with a SkyScan 1276 Micro-CT (Bruker, Billerica, MA, US) at 50-kV voltage, 70-μA current, and 40-μm voxel size. Analysis was performed using CTAn Software v1.17.9.0 (Blue Scientific Ltd., Cambridge, UK) and CTvox Software v3.3.0.0 (Blue Scientific Ltd., Cambridge, UK).

### 2.5. HE Staining

The samples fixed with paraformaldehyde were embedded in paraffin and sectioned with a slicer. The samples were dewaxed and dehydrated before HE staining. Two blinded examiners assessed lung damage using the scoring criteria described below. Intra-alveolar edema, intra-alveolar hemorrhage, and neutrophil infiltration in the tissues were scored on a scale from 0 to 4 (0, absent; 1, mild; 2, moderate; 3, severe; 4, overwhelming). As mentioned earlier, the sum of each parameter’s scores is the total histology score [17].

### 2.6. Enzyme-Linked Immunosorbent Assay (ELISA)

The lung tissues were excised and weighed, and PBS buffer was added. An ultrasonic pulverizer (Ningbo Xinzhi Biotechnology Co., Ltd., Ningbo, China) was used to lyse and homogenize the tissue. The mixture was centrifuged at 4 °C and 14,000× *g* for 20 min, and the supernatants were then transferred into new tubes. ELISA kits (Elabscience Biotechnology Co., Ltd., Wuhan, China) were used to quantify the levels of IL-β and IL-18 in tissue supernatants according to the manufacturer’s protocols. Optical density (OD) values were measured at 450 nm using a multimode plate reader (PerkinElmer Co., Ltd., Hong Kong, China).

### 2.7. Western Blot Assay

Western blotting was performed to detect protein expression in lung tissues. Total protein was extracted with RIPA lysis buffer, and the homogenates were centrifuged at 14,000× *g* for 15 min. The supernatants were transferred into new tubes, and the protein concentration was estimated using a BCA protein assay kit (Solarbio, Beijing, China). The protein concentration was determined and separated by sodium dodecyl sulfate–polyacrylamide gel electrophoresis (SDS–PAGE). The proteins were then transferred onto polyvinylidene difluoride (PVDF) membranes. The blots were blocked with 5% skim milk and incubated overnight at 4 °C with anti-NLRP3 antibody (1:1000; Abcam, Cambridge, MA, USA), anti-Caspase1 antibody (1:1000; Novus, Hong Kong, China), anti-GSDMD antibody (1:1000; Affinity, Cincinnati, OH, USA), and anti-β-actin antibody (1:10,000; Affinity, USA), then coupled with the corresponding peroxidase secondary antibody (1:5000, Affinity, Cincinnati, OH, USA) for 1 h at room temperature. The proteins were detected by a chemiluminescence (ECL) system. The expression of proteins was normalized to β-actin as a reference.

### 2.8. Immunofluorescence Staining

Paraffin sections of rat lung tissue were obtained and rinsed with 0.01 M PBS three times for 5 min each. Then, the tissue was blocked with 10% normal goat serum at 37 °C for 45 min. Suctioned excess liquid and anti-GSDMD (dilution 1:100, Affinity, USA) were added. The sections were incubated at 37 °C for 1 h and then placed in a wet box in a refrigerator at 4 °C overnight. The sections were washed with 0.01 M PBS three times for 5 min each. Goat anti-rabbit IgG-FITC (1:200) was added in the dark and incubated at 37 °C for 45 min. The secondary antibody solution was aspirated and discarded in the dark, and DAPI staining solution (2.5 mg/mL) was added and incubated at room temperature for 20 min. The sections were rinsed with 0.01 M PBS six times for 5 min each in the dark. The sections were then mounted using a fluorescence quencher in the dark and observed under a fluorescence microscope with the appropriate excitation wavelength, and images were acquired to record the experimental results. The fluorescence intensity was assessed using a Zeiss Axioskop 2 (Carl Zeiss MicroImaging, Inc., Weimar, Germany).

### 2.9. Transmission Electron Microscopy

The lung tissue was cut into 1 mm^3^ pieces, fixed with 2.5% glutaraldehyde for 24 h, and then fixed with 1% osmium tetroxide for 3 h. After being dehydrated in gradient ethanol of 50%, 70%, 90%, and 100% for 10 min, the lung tissue was embedded with resin and cut into sections with a thickness of 50–60 nm. The sections were stained with uranyl acetate and lead citrate and observed under a transmission electron microscope (HT7700, Hitachi, Tokyo, Japan). Red blood cells do not have a nucleus or organelles. Type I alveolar epithelial cells are flat, mostly located on the surface of the alveoli, with few organelles and more swallowed vesicles in the cytoplasm. Type II alveolar epithelial cells are located between type I alveolar epithelial cells, with a round nucleus and shallow cytoplasm coloration and foamy shape. Under electron microscopy, cells have osmilphilic multilamellar bodies. The cytoplasm of macrophages contains a large number of primary lysosomes, secondary lysosomes, phagocytic vesicles and phagosomes, as well as more developed Golgi complex, a small number of mitochondria and a rough endoplasmic reticulum. Other cells are also quite different from macrophages. We identified alveolar macrophages by judging the structural characteristics of these cells. Alveolar macrophages were identified by electron microscopy, pyroptotic bodies were observed, and membrane pores were further seen on the alveolar macrophage membrane, from which we deduced that it may be GSDMD membrane pores.

### 2.10. Statistical Analysis

Data are presented as the mean ± standard deviation (SD). The significance of differences was assessed with an analysis of variance (ANOVA), followed by Tukey test using GraphPad Prism 8.0.2 (GraphPad Software Inc., San Diego, CA, USA). *p* < 0.05 was considered to indicate significant differences.

## 3. Results

### 3.1. Hydrogen Partially Reversed the Increase in Lung Tissue Density and Reduced Pulmonary Inflammation

At 12 and 24 h post-injury, there was no pulmonary edema in the sham or sham + H_2_ groups, but there were diffuse exudative changes in both lungs in the burn group. The exudative changes in lung tissue in the burn + H_2_ group were alleviated. By analyzing the lung tissue density of the right upper, right middle and right lower lobes of each rat lung, we found that there was no significant difference in lung tissue density between the sham and sham + H_2_ groups. At 12 and 24 h post-injury, the lung tissue density in the burn group was significantly higher than that in the sham group (Figure 1). Hydrogen treatment partially reversed the increase in lung tissue density and reduced pulmonary inflammation.

### 3.2. Hydrogen Mitigates Histological Damage to the Lung after a Burn

At 12 and 24 h after injury, there was no obvious hyperemia, hemorrhage or inflammatory cell infiltration in the pulmonary interstitium of rats in the sham and sham + H_2_ groups, but there were obvious pathological changes in the lung tissue of rats in the burn group, including pulmonary interstitial congestion, hemorrhage and edema, severe rupture of the alveolar capillary wall and extensive infiltration of inflammatory cells. In the burn + H_2_ group, the pulmonary interstitial edema was mild, the alveolar capillary was congested with a small amount of rupture and bleeding, and the inflammatory cell infiltration was less than that in the burn group (Figure 2). Thus, hydrogen had a protective effect on ALI after burn injury.

### 3.3. Hydrogen Attenuates Burn-Induced Changes in the Expression of NLRP3 and Caspase-1

Total protein was extracted 12 and 24 h after modeling, and the expression levels of NLRP3 and caspase-1 were detected by Western blot analysis. The protein expression of NLRP3 and caspase-1 increased significantly at 12 and 24 h after the burn and showed an increasing trend. These results suggest that NLRP3 and its downstream protein caspase-1 may be involved in the pathogenesis of ALI post-burn. Hydrogen treatment partially reversed the increase in the expression of the above proteins (*p* < 0.05). There was no significant difference between the sham and sham + H_2_ groups (*p* > 0.05) (Figure 3).

### 3.4. Hydrogen Reduces Inflammation Caused by Burns

To assess the effects of burn and hydrogen treatment on the inflammatory response in rats, we evaluated the levels of IL-1β and IL-18 in the lung tissue of each group at 12 and 24 h after the establishment of the model. The expression of IL-1β and IL-18 increased significantly at 12 and 24 h after the burn and showed an increasing trend. After hydrogen treatment, the levels of IL-1β and IL-18 in the burn + H_2_ group were markedly lower than those in the burn group (*p* < 0.05) (Figure 4).

### 3.5. Hydrogen Reduced the Expression of GSDMD and AM Pyroptosis Caused by the Burn

Total protein was extracted 12 and 24 h after modeling, and the expression levels of GSDMD were detected by Western blot analysis. The protein expression of GSDMD increased significantly at 12 and 24 h post-burn injury, and hydrogen treatment partially reversed the increase in GSDMD protein expression (*p* < 0.05) (Figure 3).

At 12 and 24 h after modeling, lung tissues were collected, and the distribution of GSDMD was assessed by immunofluorescence. The distribution extent of GSDMD in lung tissue was increased 12 and 24 h post-burn, indicating that GSDMD may be involved in the occurrence of ALI post-burn. Hydrogen treatment markedly mitigated this change (*p* > 0.05). Compared with the sham group, hydrogen alone had no marked effect on GSDMD levels (*p* > 0.05) (Figure 5). Through transmission electron microscopy, we observed that the GSDMD membrane pores in AMs were increased at 12 and 24 h post-burn compared with sham rats; however, hydrogen treatment mitigated this trend (Figure 6).

## 4. Discussion

In this article, we successfully developed a 40% TBSA full-thickness burn-induced ALI rat model, which was similar to that used in previous studies, based on observations of lung histological changes, inflammatory mediator infiltration, and pulmonary microvasculature permeability.

Burns are an important global medical problem. Patients with severe burns tend to develop ALI, even without inhalation injury and infection [18]. ALI is a life-threatening disease that increases burn-related mortality. However, effective therapeutic drugs are still lacking for ALI [19]. The therapeutic effect of hydrogen has been increasingly recognized. Tong et al. observed that hydrogen has a protective effect on hindlimb ischemia–reperfusion injury in mice by reducing oxidative stress and damaging endoplasmic reticulum stress (ERS) and apoptosis [20]. In the septic model, molecular hydrogen has been proven to have multiple lung protective effects [9]. Burns can cause strong inflammation, and excessive inflammation can cause significant damage to the body. Previous studies in our group revealed that hydrogen inhalation can significantly reduce lung inflammatory cell infiltration and improve lung pathological injury in severely burned mice [21]. Therefore, this study explored the protective effect of hydrogen on ALI caused by burns and provided a new idea for the treatment of ALI.

In this study, we provide a basis for the protective effect of hydrogen on the lungs of burned rats. Studies have shown that hydrogen can improve the histological changes in the lungs caused by burns, reduce the inflammatory response of the lung, and reduce AM pyroptosis through the NLRP3/caspase-1/GSDMD pathway. We speculate that hydrogen may alleviate ALI by reducing AM pyroptosis.

We identified the pathological changes in lung tissue in burnt rats more intuitively by CT scan and evaluated the lung tissue damage caused by the burn by analyzing the density of lung tissue. For simple burns, excessive inflammation caused by burns can immediately cause the activation of endogenous AMs and neutrophils in circulation, thus increasing pulmonary density [22]. The main pathological features of ALI are damage to the pulmonary microvascular endothelium and alveolar epithelium, an increase in pulmonary microvascular permeability and the exudation of protein-rich fluid from the alveolar cavity, which leads to pulmonary edema. Therefore, the lung tissue density of ALI induced by burns will increase. Yang et al. observed an increase in lung tissue density in burned rats and confirmed that CT-scan density analysis can be used as a method to evaluate lung injury caused by burn-blast injury [23]. In this study, it was observed that pulmonary density in the burn group increased significantly at 12 and 24 h post-burn, and the density at 24 h was higher than that at 12 h. Hydrogen treatment can significantly improve the increase in lung tissue density in burned rats, indicating that hydrogen inhibits the inflammatory response caused by burns.

Burning is a complex pathophysiological process, and severe burns can cause a variety of inflammatory reactions. As inflammatory-initiating cytokines, IL-1β and IL-18 spread injury signals and trigger inflammatory cascade reactions. Our study showed that the expression of IL-1β and IL-18 was markedly increased at 12 and 24 h post-burn injury. The results are consistent with the results of Han et al. [24]. Hydrogen can significantly reduce the secretion of IL-1β and IL-18 and reduce pulmonary inflammation. NLRP3 is called the secretion platform of proinflammatory cytokines [25]. Roohi et al. observed the activation of NLRP3 in white adipose tissue of burn patients and found that NLRP3 has an anti-browning effect [26]. Han et al. observed the activation of NLRP3 in the lung tissue of burned rats [24]. Roohi et al. observed the upregulated expression of NLRP3, IL-1β and IL18 in skin tissue post-burn and found that NLRP3 has a protective effect on burn wound healing [25]. In this study, it was observed that the expression of NLRP3 and its downstream caspase-1 increased at 12 and 24 h after burn injury, which confirmed that burn injury activated the NLRP3/caspase-1 pathway. However, Mile et al. found that the acute inflammatory response of NLRP3 knockout mice increased after burn, but the survival rate increased [27]. GSDMD, as a downstream protein of the NLRP3/caspase-1 pathway, is also activated after burn. At the same time, we observed an increase in the membrane pores of AMs after burn injury, which corresponded to the activation of GSDMD. This result indicates that the burn aggravates AM pyroptosis. Notably, hydrogen can inhibit the NLRP3/caspase-1/GSDMD pathway, reduce AM membrane pore formation, reduce pyroptosis, and reduce the release of inflammatory factors; thus, part of the reason hydrogen reduces lung tissue density and burn-induced ALI may be due to reduced cell pyroptosis. A study suggests that melatonin exerts protective roles on LPS-induced ALI and pyroptosis by inhibiting the NLRP3-GSDMD pathway [28]. GSDMD may regulate NLRP3 inflammasome involved in Benzo(a)pyrene (B(a)P) induced inflammatory damage in alveolar epithelial cells (A549 cells) [29]. Honokiol (HKL) inhibits the expression levels of NLRP3, CASP1, and GSDMD and could attenuate the pathological injury in LPS-induced ALI rats [30]. These studies support that inhibition of NLRP3/caspase-1/GADMD activation could reduce lung injury. Interestingly, we also observed that the expression of GSDMD protein was markedly higher than that of NLRP3/caspase-1, and we speculated that NLRP3/caspase-1 may not be the only pathway acting on GSDMD in the case of burn. The specific mechanism needs to be further explored.

However, the current study has some limitations. First, the effect of hydrogen on other inflammatory complexes, such as NLRP1, NLRC4 and AMI2, is unknown. Second, we did not establish a burn-induced ALI model in NLRP3−/− rats, so the blocking degree of hydrogen on the NLRP3 inflammatory body was not well elucidated.

In conclusion, the results show that hydrogen can inhibit AM pyroptosis induced by burn-induced ALI and protect the lung through the NLRP3/caspase-1/GSDMD pathway.

## Figures and Tables

**Figure 1 jpm-13-00377-f001:**
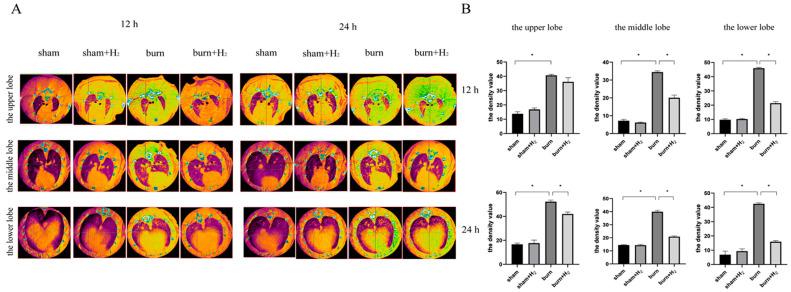
Hydrogen partially reversed the increase in lung tissue density and reduced pulmonary inflammation. The pulmonary inflammation was evaluated by micro-CT (**A**). Average density values for each lobe at 12 and 24 h after injury were analyzed (**B**) (*n* = 6 rats per group). * *p* < 0.05.

**Figure 2 jpm-13-00377-f002:**
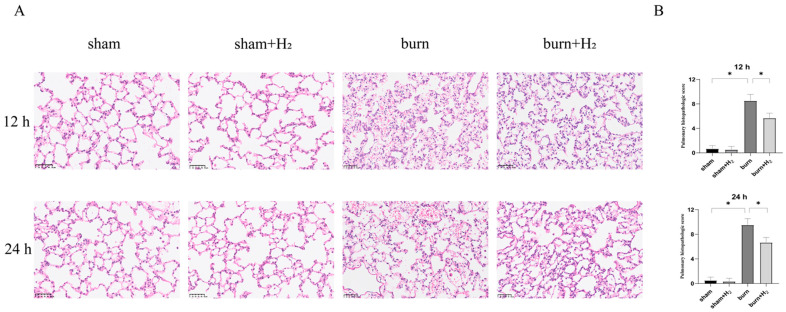
Effects of burns and hydrogen on pulmonary histopathology. Rats were sacrificed 12 h and 24 h after sham or burn operation, and lung tissues were harvested for HE staining (**A**). Lung injury was scored by visualizing the morphological structure (**B**) (*n* = 6 rats per group). * *p* < 0.05.

**Figure 3 jpm-13-00377-f003:**
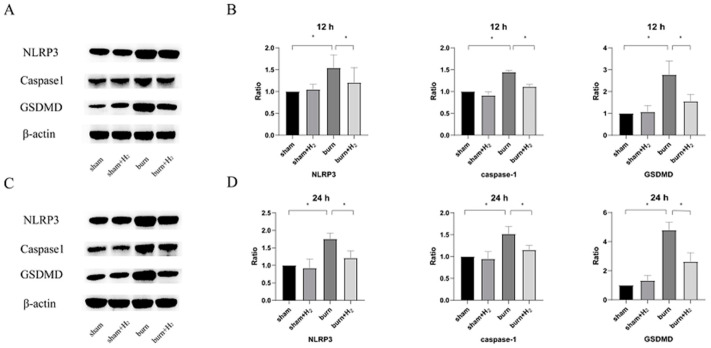
Hydrogen attenuates burn-induced changes in the expression of NLRP3, caspase-1 and GSDMD at 12 (**A**) and 24 h (**C**) after the burn. The expression levels of NLRP3, caspase-1 and GSDMD were detected by Western blot analysis. Quantitative analysis of NLRP3, caspase-1 and GSDMD (**B**,**D**) levels, shown as the ratio of each protein band density to that of β-actin (*n* = 6 rats per group). * *p* < 0.05.

**Figure 4 jpm-13-00377-f004:**
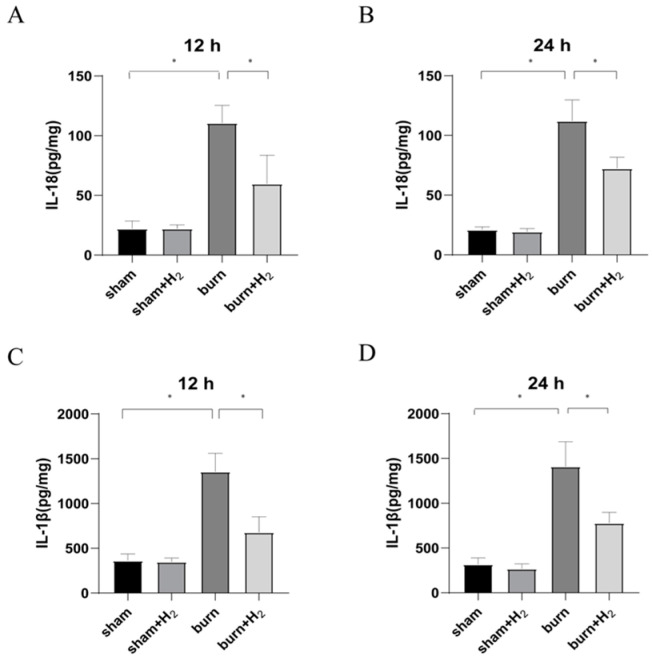
Effects of hydrogen treatment after burn injury on the inflammatory response in rat lung tissues. The levels of IL-18 (**A**,**B**) and IL-1β (**C**,**D**) in lung tissue in each group were measured with ELISA kits (*n* = 6 rats per group). * *p* < 0.05.

**Figure 5 jpm-13-00377-f005:**
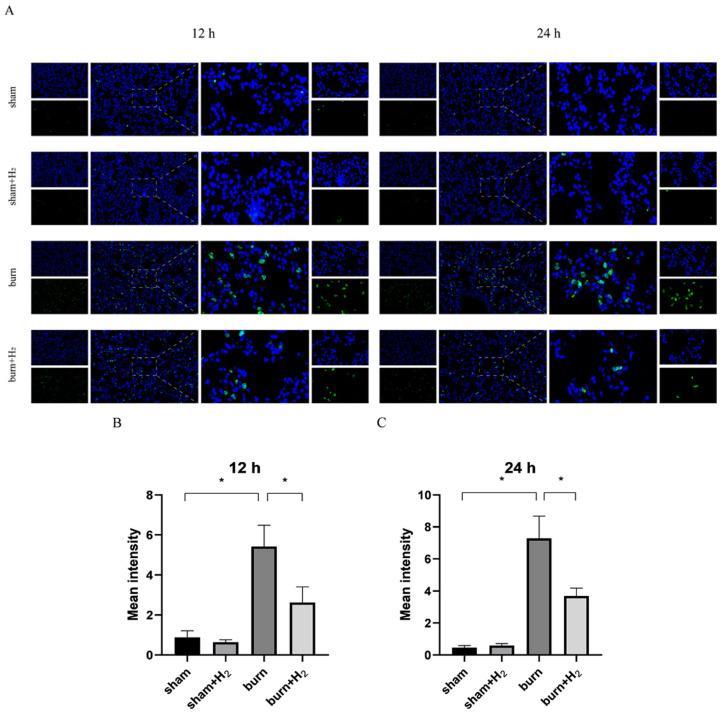
Hydrogen reduced the expression of GSDMD caused by burn. The distribution of GSDMD was detected by immunofluorescence at 12 and 24 h (**A**). The density of GSDMD was analyzed (**B**,**C**) (*n* = 4 rats per group). * *p* < 0.05.

**Figure 6 jpm-13-00377-f006:**
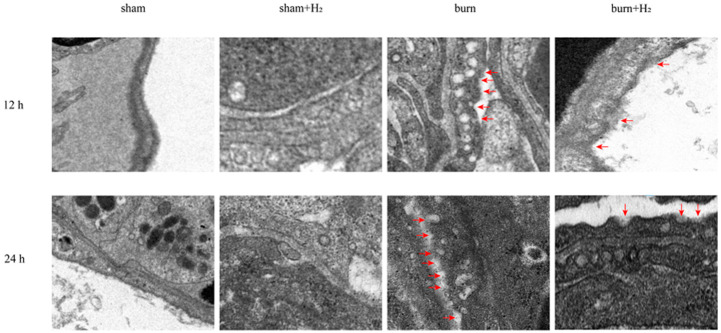
Hydrogen reduced AM pyroptosis caused by the burn in rats. The GSDMD membrane pores in AMs were observed by transmission electron microscopy. Red arrowhead: membrane pores. (*n* = 4 rats per group).

## Data Availability

The datasets used and analysed in the present research are available from the corresponding author on reasonable request.

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
