# Peer review of "Effect of Hydrogen on AM Pyroptosis Induced by Severe Burns in Rats"

_jpm, 2023, doi:10.3390/jpm13030377_

Round 1

Reviewer 1 Report

It is interesting to see that inhalation of hydrogen again show some measurable modifying effects in an experimental model; this time for ALI (acute lung injury). The lack of toxicity of hydrogen is noted.                       Major comments / questions: (1) how did you decide upon the dose and exposure time period of hydrogen? Please comment. (2) CT-data is of course very interesting, as well as histology. The laboratory work impressed me. Did you do any experiments on wet/ dry ratio of lung tissue? It is a much-used parameter when discussing degree of lung injury.  

Minor comments: In Fig 1, the text for Y-axis right panel is missing. In the reference list, ref no 18 seems to be incomplete. 

Author Response

Dear reviewer:

Thank you for spending time in reviewing our manuscript and providing us with a list of constructive comments.

We have carefully considered the suggestion of Reviewer and make some changes. We have tried our best to improve and made some changes in the manuscript.

Major comments / questions: (1) how did you decide upon the dose and exposure time period of hydrogen? Please comment.

Several literature has demonstrated a protective effect of inhalation of 2% H2 at 1 and 6 hours after injury.【Zhuang X, Yu Y, Jiang Y, Zhao S, Wang Y, Su L, Xie K, Yu Y, Lu Y, Lv G. Molecular hydrogen attenuates sepsis-induced neuroinflammation through regulation of microglia polarization through an mTOR-autophagy-dependent pathway. Int Immunopharmacol. 2020 Apr;81:106287. doi: 10.1016/j.intimp.2020.106287. Epub 2020 Feb 10. PMID: 32058932.】【Qin Chao, et al. Effects of hydrogen on the lung damage of mice at early stage of severe burn. China J Burns, 2017. 33(11): p.682-687.】At 1 and 6 hours post-injury, rats in the sham + H2 and burn + H2 groups inhaled hydrogen for 1 hour, while rats in the sham and burn groups were in the same environment and inhaled air.

(2) CT-data is of course very interesting, as well as histology. The laboratory work impressed me. Did you do any experiments on wet/ dry ratio of lung tissue? It is a much-used parameter when discussing degree of lung injury.  

Our experimental group has studied the dry-wet ratio in previous experiments, please refer to the article【Qin Chao, et al. Effects of hydrogen on the lung damage of mice at early stage of severe burn. China J Burns, 2017. 33(11): p.682-687.】

Minor comments: In Fig 1, the text for Y-axis right panel is missing. In the reference list, ref no 18 seems to be incomplete. 

It has been modified.

We hope the revised manuscript is now acceptable to you. If not, we are glad to receive any further feedback which we shall continue to apply our best effort to address.

Reviewer 2 Report

This research was to study the possible positive effects of hydrogen on burn-induced lung injury and the effects of hydrogen on AM pyroptosis during acute lung injury (ALI) induced by burns in rats.

Methods: Histological changes in rat lungs in vivo were evaluated by micro-CT, and histological changes in isolated lungs were evaluated by H&E staining. The expressions of leucine rich repeat (LRR) and pyrin domain (PYD) containing protein 3 (NLRP3), caspase1 and Gasdermin-D (GSDMD) were analysed by Western blot. The expression of GSDMD was also quantified by IF to evaluate the levels of lung inflammation and pyroptosis. The level of inflammation was assessed by enzyme-linked immunosorbent assay (ELISA) of  IL-β and IL-18. Pyroptosis was observed by transmission electron microscopy.

Findings: Hydrogen protected from lung injury based upon loss of air-filled lungs by CT chest, H&E evidence of injury (composite score), and decreased inflammation based on IL-beta and IL-18.  Pyroptosis in this model was evoked on the strength of increased NLRP3, GSDMD and caspase 1, and these endpoints were mitigated in the presence of 2% inhaled hydrogen.  TEM showed structures potentially consistent with GSDMD pores in burned rats, which are hypothesized to provide support for the mitigation of pyroptosis pathway by hydrogen.

Overall, the findings support hydrogen evoked protection from pyroptosis in AMs through the NLRP3/caspase-1/GSDMD pathway in a rodent model of burn injury.

Previous work has demonstrated that hydrogen can improve the histological changes of burn injury in rodent lungs and reduce the inflammatory responses, so the new findings in this manuscript are localized to the potential role of pyroptosis.  

Major concerns:

The choice of pulmonary antigens to detect (NLRP3, GSDMD, LRR) should be discussed.  Why these endpoints? Were others considered?

The title of the paper alludes to AM pyroptosis, but most of the studies were performed on lung homogenates, which includes potentially 40+ cell types.  Why not include some studies of lavaged macrophages?  If the hypothesis that pyroptosis specifically in alveolar macrophages is mitigated by H2, the endpoints of at least Caspase 1, NLRP3 and GSDMD quantitation by WB could be assessed in lavaged AMs. 

The statistical descriptions are inadequate.  What is the total number of rats used for all experiments, and then each experiment?  Were data normally distributed or not?  If ANOVA was used to compare 4 groups, what post hoc tests were employed? 

Variance of WB values in the sham groups should be shown in the graphs.

Were H&E sections scored by evaluators blinded to the treatment group?

For figure 5, does mean density imply intensity and/or area of the IF images with immunospecific uptake?  Exactly what was quantified?

The statement on page 5 “obvious inflammatory reactions in lung tissue could be seen” on microCT should be changed.  Inflammation cannot be determined by CT; infiltrates can.  The authors measured air or fluid densities in the CT scans, not inflammation.

How were areas to evaluate for pores selected for comparing the 4 groups?  Was any quantitative comparison attempted? There is no description of how macrophages were identified.

For Figure 6, the authors say they are evaluating GSDMD membrane pores.  In fact, the TEMs point to relatively non-specific evaginations. This fact should be acknowledged.

Minor concerns:

It is not entirely clear when/how 2% hydrogen was administered.  It appears that it was given for 1 hour duration at both 1 and 6 hours after the burn.  This point should be clarified. 

Were lungs fixed inflated?  If so, how was this accomplished?

Author Response

Dear reviewer:

Thank you for spending time in reviewing our manuscript and providing us with a list of constructive comments.

We have carefully considered the suggestion of Reviewer and make some changes. We have tried our best to improve and made some changes in the manuscript.

Major concerns:

1.The choice of pulmonary antigens to detect (NLRP3, GSDMD, LRR) should be discussed.  Why these endpoints? Were others considered?

Caspase-1-mediated pyroptosis is the first classical pyroptosis pathway discovered, so we wanted to investigate whether burns cause pyroptosis and the effects of hydrogen intervention.

2.The title of the paper alludes to AM pyroptosis, but most of the studies were performed on lung homogenates, which includes potentially 40+ cell types.  Why not include some studies of lavaged macrophages?  If the hypothesis that pyroptosis specifically in alveolar macrophages is mitigated by H2, the endpoints of at least Caspase 1, NLRP3 and GSDMD quantitation by WB could be assessed in lavaged AMs. 

Many thanks to the reviewers for their valuable and constructive comments. We first used lung tissue for experiments, verified the activation of NLRP3-Caspase-1-GSDMD in lung tissue after burn by WB, ELISA and IF, identified lung macrophages by TEM, observed pyroptotic bodies, and further saw membrane pores on the alveolar macrophage membrane, from which we deduced that it may be pyroptosis of alveolar macrophages. Of course, more definitive conclusions require further research, and your comments have brought us new ideas, and we will incorporate your valuable ideas in future studies to conduct further experiments using alveolar lavage fluid.

3.The statistical descriptions are inadequate.  What is the total number of rats used for all experiments, and then each experiment?  Were data normally distributed or not?  If ANOVA was used to compare 4 groups, what post hoc tests were employed?

 The number of rats in each group was 28, and the total number of rats used in the experiment (plus the rats that died due to modeling) was 120. The data conform to a normal distribution. The significance of differences was assessed with an analysis of variance (ANOV A) followed by Tukey test using the GraphPad Prism 8.0.2. P< 0.05 was considered to indicate significant differences.

4.Variance of WB values in the sham groups should be shown in the graphs.

We plotted the ratio of the grayscale values of the other groups to the Sham group. Consider the sham group as 1.

5.Were H&E sections scored by evaluators blinded to the treatment group?

Yes, two blinded examiners assessed lung damage using the scoring criteria.

6.For figure 5, does mean density imply intensity and/or area of the IF images with immunospecific uptake?  Exactly what was quantified?

We used imageJ software to measure fluorescence intensity to quantify IF, which has been perfected as depicted in the figure.

7.The statement on page 5 “obvious inflammatory reactions in lung tissue could be seen” on microCT should be changed.  Inflammation cannot be determined by CT; infiltrates can.  The authors measured air or fluid densities in the CT scans, not inflammation.

The inappropriate description has been modified.

8.How were areas to evaluate for pores selected for comparing the 4 groups?  Was any quantitative comparison attempted? There is no description of how macrophages were identified.

Red blood cells do not have a nucleus and organelles. Type I alveolar epithelial cells are flat, mostly located on the surface of the alveoli, with few organelles and more swallowed vesicles in the cytoplasm. Type II alveolar epithelial cells are located between type I alveolar epithelial cells, with a round nucleus and shallow cytoplasm coloration and foamy shape. Under electron microscopy, cells have osmilphilic multilamellar bodies. The cytoplasm of macrophages contains a large number of primary lysosomes, secondary lysosomes, phagocytic vesicles and phagosomes, as well as more developed Golgi complex, a small number of mitochondria and a rough endoplasmic reticulum. Other cells are also quite different from macrophages. We identify alveolar macrophages by judging the structural characteristics of these cells.

9.For Figure 6, the authors say they are evaluating GSDMD membrane pores.  In fact, the TEMs point to relatively non-specific evaginations. This fact should be acknowledged.

Alveolar macrophages were identified by electron microscopy, pyroptotic bodies were observed, and membrane pores were further seen on the alveolar macrophage membrane, from which we deduced that it may be GSDMD membrane pores. We tried to quantify the comparison of GSDMD pores, but it was a little difficult due to limitations in the clarity of tissue TEM.

Minor concerns:

1.It is not entirely clear when/how 2% hydrogen was administered.  It appears that it was given for 1 hour duration at both 1 and 6 hours after the burn.  This point should be clarified. 

Several literature has demonstrated a protective effect of inhalation of 2% H2 at 1 and 6 hours after injury【Zhuang X, Yu Y, Jiang Y, Zhao S, Wang Y, Su L, Xie K, Yu Y, Lu Y, Lv G. Molecular hydrogen attenuates sepsis-induced neuroinflammation through regulation of microglia polarization through an mTOR-autophagy-dependent pathway. Int Immunopharmacol. 2020 Apr;81:106287. doi: 10.1016/j.intimp.2020.106287. Epub 2020 Feb 10. PMID: 32058932.】【Qin Chao, et al. Effects of hydrogen on the lung damage of mice at early stage of severe burn. China J Burns, 2017. 33(11): p.682-687.】

In a sealed plexiglass box with an inlet and an outlet, a hydrogen generator was used to generate hydrogen, and the hydrogen and air were mixed by a gas flowmeter and entered the box at a rate of 4 L/min. The oxygen volume fraction in the box was 21%, and the hydrogen volume fraction was maintained at 2% with a hydrogen detector. A layer of calcium lime at the bottom of the box was used to absorb the carbon dioxide produced by the rats during breathing. At 1 and 6 hours post-injury, rats in the sham + H2 and burn + H2 groups inhaled hydrogen in this environment for 1 hour, while rats in the sham and burn groups were in the same environment and inhaled air.

2.Were lungs fixed inflated?  If so, how was this accomplished?

The lungs were not fixed inflated.

We hope the revised manuscript is now acceptable to you. If not, we are glad to receive any further feedback which we shall continue to apply our best effort to address.

Happy Lantern Festival to you!

Round 2

Reviewer 2 Report

From my standpoint, the authors have responded thoughtfully to the comments. 

The paper is now acceptable for publication in my opinion.